# WRITING IN THE MARGINS: BETTER INFERENCE PATTERN FOR LONG-CONTEXT RETRIEVAL

## ABSTRACT

In this paper, we introduce Writing in the Margins (WiM), a new inference pattern for Large Language Models designed to optimize the handling of long input sequences in retrieval-oriented tasks. This approach leverages the chunked prefill of the key-value cache to perform segment-wise inference, which enables efficient processing of extensive contexts along with the generation and classification of intermediate information ("margins") that guide the model towards specific tasks. This method increases computational overhead marginally while significantly enhancing the performance of off-the-shelf models without the need for fine-tuning. Specifically, we observe that WiM provides an average enhancement of 7.5% in accuracy for reasoning skills (HotpotQA, MultiHop-RAG) and a 30.0% increase in the F1-score for aggregation tasks (CWE). Additionally, we show how the proposed pattern fits into an interactive retrieval design that provides end-users with ongoing updates about the progress of context processing, and pinpoints the integration of relevant information into the final response. We release our implementation of WiM using Hugging Face Transformers library at <anonymised URL>.

## 1 INTRODUCTION

The performance of Large Language Models (LLMs) tends to deteriorate when processing extensive inputs, a limitation linked directly to their fixed context window and attention mechanisms (Li et al., 2024; Liu et al., 2024). In particular, LLMs struggle with tasks involving long contexts, especially when the relevant information is embedded in larger volumes of text (Bai et al., 2024; Shaham et al., 2023). Recent research thus highlights the importance of improving model capabilities to handle more extensive datasets without losing accuracy or requiring exponential increases in computational resources.

There have been various attempts to extend the usable context window of LLMs, such as sparse attention (Tworkowski et al., 2023; Chen et al., 2024; Mohtashami & Jaggi, 2023), length extrapolation (Dai et al., 2019; Su et al., 2023; Peng et al., 2024), and context compression (Ge et al., 2024; Mu et al., 2023). Concurrently, the field has witnessed the rise of sophisticated prompting strategies like Chain of Thought (CoT) and related structured reasoning methods (Wei et al., 2022; Yao et al., 2023; Besta et al., 2024). These approaches have significantly enhanced LLMs' ability to tackle complex tasks by systematically guiding the reasoning process through predefined structural patterns.

Our work bridges the gap between efficient transformers architecture research and development of new prompting strategies. Specifically, we identify a novel key-value (KV) cache aware reasoning pattern for existing off-the-shelf long context window LLMs in scenarios typical of retrieval-oriented tasks, where the context is substantial and the instructional prompt is comparatively short. We begin by recognizing that long-context prompts are commonly prefilled in the KV cache segment-wise in a process known as chunked prefill. From this insight, we introduce an inference pattern called Writing in the Margins (WiM), which concurrently generates query-based extractive summaries at each step of the prefill that are subsequently reintegrated at the end of the computation. We term these intermediate outputs "margins", drawing inspiration from the practice of making margin notes for improved comprehension of long contexts in human reading. Using methodologies similar to "scratchpad" techniques, which meticulously record step-by-step calculations, we incorporate margin notes into the final segment predictions. We show that this technique, which adds only minimal

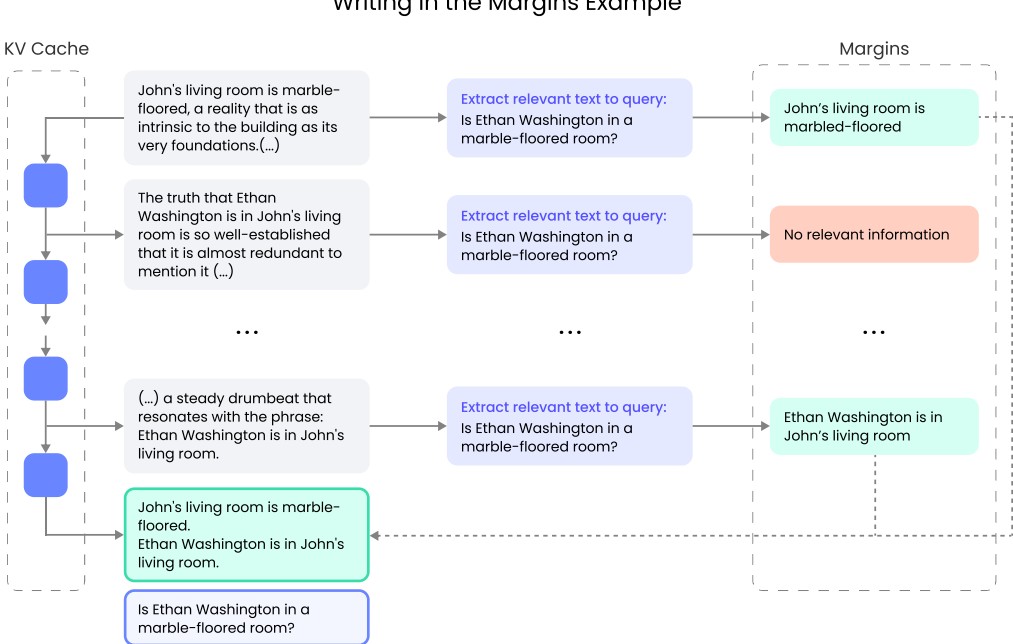

Figure 1: **Writing in the Margins inference pattern.** Prefilling KV cache by segments allows to both process the context segment by segment and generate intermediate extractive summaries which can improve the final prediction.

additional computation, significantly enhances long context comprehension. The WiM pattern can also provide end-users with real-time insights into computational progress through streamed margin notes, which ultimately help make AI decisions more transparent and explainable. This can enable users to (1) pinpoint the location of essential information and (2) reduce computational load by exiting early if the provided information satisfactorily addresses the query.

In Figure 1, we provide an illustrative example of WiM inference, which we encourage readers to reference as a practical demonstration to complement the formal algorithm description that will be presented in the following sections.

Our main contributions are as follows:

- We introduce a new inference pattern, Writing in the Margins (WiM), which achieves better performance on long-context window tasks with a relatively minor increase in computational cost.
- We demonstrate the application of WiM within an interactive long context retrieval setup, effectively increasing the transparency of the process and reducing the first response latency.
- We provide an implementation of this inference pattern using the Hugging Face Transformers library.

## 2 WRITING IN THE MARGINS

### 2.1 CHUNKED PREFILL

Typically, the process of inference for generative LLMs consists of two principal phases: the prefill phase and the decoding phase. When an LLM is requested to prefill a substantial prompt—in the range of hundreds of thousands of tokens—it is common practice to prefill the KV cache in chunks

Table 1: **Chunked Prefill.** Example of how the attention mask is set across different chunks during prefill iterations (first chunk on the left, second chunk on the right). Each new chunk needs to retain causality while attending to all previous chunks. Chunked prefill is mathematically equivalent to prefill without chunking.

|      | K0 | K1 | K2 | K3 |      | K0 | K1 | K2 | K3 | K4 | K5 | K6 | K7 |
|------|----|----|----|----|------|----|----|----|----|----|----|----|----|
| Q0   | 1  | 0  | 0  | 0  | Q4   | 1  | 1  | 1  | 1  | 1  | 0  | 0  | 0  |
| Q1   | 1  | 1  | 0  | 0  | Q5   | 1  | 1  | 1  | 1  | 1  | 1  | 0  | 0  |
| Q2   | 1  | 1  | 1  | 0  | Q6   | 1  | 1  | 1  | 1  | 1  | 1  | 1  | 0  |
| Q3   | 1  | 1  | 1  | 1  | Q7   | 1  | 1  | 1  | 1  | 1  | 1  | 1  | 1  |

(Agrawal et al., 2024). This method is known as chunked prefill and is supported by many inference frameworks, including vLLM (vLLM, 2024).

Chunked prefill divides the prompt into fixed-size chunks to populate the KV cache at each layer of the Transformer model (Vaswani et al., 2017). The rationale for chunked prefill is to reduce overall memory usage, as the quadratic memory complexity of the attention mechanism during prefilling can be prohibitive for larger prompts. By splitting a prompt of length $L$ into $N$ chunks, each of size $K$, where $N = L/K$, the overall memory complexity of prefilling is reduced from $O(L^2)$ to $O(LK)$. The attention mask must be adjusted to allow each new chunk to attend to all tokens in the previous chunks while maintaining the causal structure only for the new chunk, as illustrated in Table 1.

Our work exploits the chunked prefill mechanism to generate intermediate "margins" that can then be appended to the prompt to better guide the model toward performing a specific task.

## 2.2 WRITING IN THE MARGINS

Consider a prompt $P$, composed of a context $C$, and an instruction $I$. Prefilling a decoder-only transformer model $T$ directly with the entire prompt $T(P)$ is computationally inefficient when the prompt is long. Moreover, as shown in Liu et al. (2024), processing the entire prompt in one go can lead to mid-sequence forgetting.

To make this process more efficient, we implement the prefill technique described in the previous paragraph, where the context C is divided into $N$ segments; i.e., $C = c_1 + c_2 + ... + c_N$. For the first segment, the model $T$ operates on chunk $c_1$, resulting in output that includes past key values $pkv_1$. The model continues onto the second segment with the $pkv_1$ cached, i.e., $T(pkv_1, c_2)$, effectively emulating the scenario of processing $T(c_1 + c_2)$ in one step. As the procedure progresses, each sequential chunk, $c_k$, is processed with prefilled past key values, noted as $T(pkv_{[1..k-1]}, c_k)$, mimicking an uninterrupted run of $T$ on C.

The Writing in the Margins (WiM) strategy addresses potential mid-sequence forgetting issues by appending an extractive instruction $I_A$ to each chunk, enhancing chunk-specific outputs. It transforms each step into $T(pkv_{[1..k-1]}, c_k + I_A)$, where the instruction $I_A$ is embedded alongside each context chunk, then dropped from the KV cache before the next chunk prefilling. The instruction $I_A$ is closely related to $I$ - the model is asked to copy over all relevant to $I$ information.

Intermediate outputs from each chunk are referred to as margin notes $M_i$, cumulatively forming $N$ notes, described as $M = M_{[1..N]}$. Unhelpful notes, perhaps irrelevant to the instruction, are discarded, enhancing the final contextual construct to $C + M + I$, positioned advantageously towards the end to minimize mid-sequence memory loss. Intuitively, the model is allowed to use relevant intermediate predictions while answering the final query.

To summarize, we modify the chunked prefill algorithm by adding extra decoding steps (green in Table 2). Most of these steps can be efficiently batched with the original prefill steps. The query-relevant information extracted from these steps is then added at the end of the context but before the instruction (see Appendix A for a pseudocode example).

Table 2: **Batching Chunked Prefill Steps with WiM margin generation.** The inference for generative LLMs consists of two principal phases: the prefill phase (†) and the decoding phase (‡). The WiM algorithm adds extra decoding steps that mostly can be batched with chunked prefill steps. We keep margin notes $M_i$ produced in extra steps (green) as plain text. We then prefill the model $T$ with all relevant notes $M_{[1..N]}$ before the final instruction $I$.

| step | Chunked Prefill | WiM | keep |
|---|---|---|---|
| 1 | $T(\emptyset, c_1)^\dagger$ | $T(\emptyset, c_1)^\dagger$ | $\text{pkv}_{[1]}$ |
| 2 | $T(\text{pkv}_{[1]}, c_2)^\dagger$ | $T(\text{pkv}_{[1]}, c_2)^\dagger$ | $\text{pkv}_{[1..2]}$ |
| | | $T(\text{pkv}_{[1]}, I_A)^{\dagger\ddagger}$ | $M_1$ |
| $\vdots$ | $\vdots$ | $\vdots$ | $\vdots$ |
| N | $T(\text{pkv}_{[1..N-1]}, c_N)^\dagger$ | $T(\text{pkv}_{[1..N-1]}, c_N)^\dagger$ | $\text{pkv}_{[1..N]}$ |
| | | $T(\text{pkv}_{[1..N-1]}, I_A)^{\dagger\ddagger}$ | $M_{N-1}$ |
| N + 1 | | $T(\text{pkv}_{[1..N]}, I_A)^{\dagger\ddagger}$ | $M_N$ |
| N + 2 | $T(\text{pkv}_{[1..N]}, I)^{\dagger\ddagger}$ | $T(\text{pkv}_{[1..N]}, M_{[1..N]} + I)^{\dagger\ddagger}$ | |

Table 3: **Datasets** We curated four datasets to evaluate long context window LLMs. Each set consists of 100 examples, generated either using RULER code (†) or by subsampling the longest examples from the original benchmark data (♣).

| skill type | benchmark name | context length (tokens) | # examples |
|---|---|---|---|
| I | MultiHop-RAG ♣ (Tang & Yang, 2024) | 13-32k | 100 |
| I | HotpotQA [†] (Yang et al., 2018) | 16k/ 32k /64k | 100/ 100/ 100 |
| II | SQuAD [†] (Rajpurkar et al., 2018) | 16k/ 32k/ 64k | 100/ 100/ 100 |
| III | CWE [†] (Hsieh et al., 2024) | 64k | 100 |

## 3 EXPERIMENTAL SETUP

### 3.1 DATASETS

Following the RULER task categories (Hsieh et al., 2024), we measure the performance of an inference pattern on three types of skills: **(I) Multi-Hop Reasoning**, **(II) Needle Retrieval/ Single-Hop Reasoning**, and **(III) Aggregation**. Table 3 presents the curated long context datasets used to benchmark all LLMs:

In the following paragraph, we briefly introduce the benchmarks used in each category and describe our curating rationale.

I. **Multi-Hop QA** The task aims to check the behavior of tracing entities with multi-hop connections based on the HotPotQA and MultiHop-RAG benchmarks (Tang & Yang, 2024; Yang et al., 2018). We used the RULER codebase[1] to generate a subset of 100 examples based on HotPotQA - a multi-hop queries sourced from Wikipedia articles. Following RULER, we simulated long context retrieval scenarios by generating examples in three length variants: $16k$, $32k$, $64k$. We also selected the 100 longest examples in the range of $13k$-$33k$ tokens from MultiHop-RAG - a large collection of multi-hop queries based on English news articles.

---

[1]https://github.com/hsiehjackson/RULER

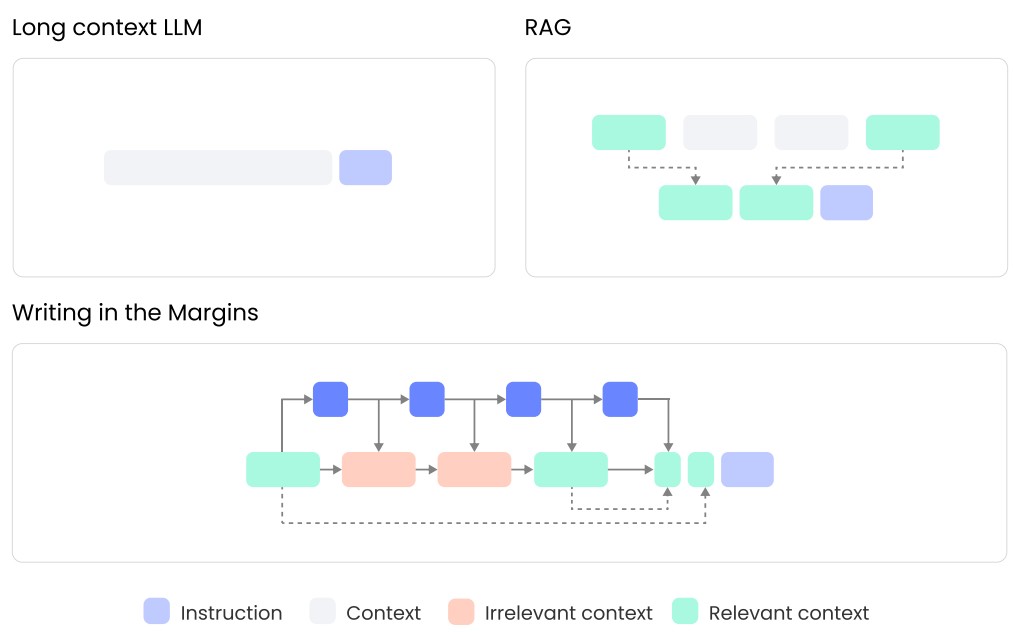

Figure 2: **Design Comparison.** Design Comparison. Three inference designs for long contexts: (Top Left) Long Context LLM: Feeds entire context to the model without segmentation. (Top Right) Retrieval-Augmented Generation (RAG): Uses a retrieval method (e.g., cosine similarity) to select segments, which are then concatenated with task instructions for the model. (Bottom) Writing in the Margins (WiM): Divides and processes context by segments, prompting the model to generate auxiliary information from each, which is classified and potentially incorporated before the task description.

II. **Needle Retrieval/ Single-Hop Reasoning** In the context of a long context window, the Needle Retrieval and Single-Hop QA task can be jointly seen as a kind of filter benchmark, where the task is to filter irrelevant content and either copy or transform the relevant information. We used the RULER code to generate examples based on SQuAD (Rajpurkar et al., 2018) in three context length variants: $16k$, $32k$ and $64k$, collecting 100 datapoints in each variant.

III. **Aggregation** This task evaluates a model's ability to aggregate relevant information across a long-range context, using the Common Words Extraction (CWE) benchmark (Hsieh et al., 2024). In this benchmark, word distribution numbers are fixed with the sequence length, using 100 examples averaging 64k tokens each. Common words appear 500 times, while uncommon words appear no more than 50 times. Task instructions were adapted to include word occurrence counts to facilitate segment aggregation.

### 3.2 LONG CONTEXT WINDOW LLMS

We selected seven off-the-shelf models that officially support context windows up to $128k$ tokens: Phi-3-small-128k-instruct (Abdin et al., 2024), Qwen2-7B-Instruct (Yang et al., 2024), Meta-Llama-3.1-8B-Instruct Dubey et al. (2024), Phi-3-medium-128k-Instruct (Abdin et al., 2024), Palmyra-4-Chat-128K (Writer's proprietary model), Meta-Llama-3.1-70B-Instruct Dubey et al. (2024), Qwen2-72B-Instruct (Yang et al., 2024).

In all experiments, we used half precision models with identical sampling parameters — specifically, a temperature setting of $0.0$ and $2k$ maximum new tokens. We used $0$-shot prompts for all benchmarks. In MultiHop-RAG, HotPotQA and SQuAD experiments, we applied the same model-

independent prepossessing step: we used nltk (Bird et al., 2009) to split the context into sentences, then grouped them in segments no longer than $4096$ tokens. This resulted in $4 - 16$ margin notes per datapoint. In CWE, where the datapoints contain only numbered words, we exchanged nltk for naive words split by space and used $8192$ segment length, which gave on average $8$ margins per sample. We chose to count tokens using GPT-4 tiktoken tokenizer[2] since this choice does not favour any of the evaluated models' tokenizers.

In each case, we measured the relative differences of WiM pattern scores with respect to the following two baselines:

- **Long Context LLM (LLM)** - all context without segmentation is fed to the LLM.
- **Retrieval Augmented Generation (RAG)** - segments are selected based on a retriever (ex. cosine similarity between vector representations of the query and the segment), then all selected segments and the task instruction are concatenated and fed to an LLM.

In order to make the results more comparable, we replaced the retriever in RAG with the classifier used in WiM. We expect the RAG results to be lower in the real RAG systems (especially for longer segment lengths), as vectorization is a form of lossy compression. All three inference patterns, including WiM, are presented in Figure 2.

## 3.3 EVALUATION

### 3.3.1 PREDICTION

In the margin accumulation step, in order to distinguish the content of the margins from the original context, and to maintain the document's logic and structure, we explicitly named the writing-in-the-margins strategy by reformatting the margins the following way:

```text
I asked my assistant to read and analyse the above content page by page
    ↪ to help you complete this task. Those are margin notes left on each
    ↪ page:
```text
Page 0:
QUERY: {query}
ANSWER: {M_i}
Page 1:
QUERY: {query}
ANSWER: {M_j}
...
```

The output is appended at the end of the final prompt. Full prompts are shown in Appendix C.

### 3.3.2 SCORING

We used the same 3-shot prompt with GPT-4-turbo (OpenAI, 2023) and greedy sampling to evaluate models' accuracy in HotpotQA, MultiHop-RAG and SQuAD benchmarks. For the CWE benchmark we adjusted the prompt and examples to calculate precision (P), recall (R) and F1-score. Both prompts are shown in Appendix C.

## 4 RESULTS

### 4.1 MULTI-HOP REASONING

Detailed results for all experiments are presented in Table 4. Notably, for almost all evaluated models, WiM improves multi-hop reasoning abilities, on average giving a $7.5\%$ boost with respect to the Long Context LLM inference and $9\%$ with respect to RAG. The most significant performance boost is observed in smaller models — replacing a vanilla Phi-3-small-128k-instruct inference with WiM leads to $19\%$ improvement in MultiHop-RAG benchmark and $12\%$ in HotpotQA.

---

[2]`https://github.com/openai/tiktoken`

Table 4: **Main Results** We show results for seven models and four benchmarks, using accuracy for all but CWE, where precision, recall, and F1-score were used. Aggregated results indicate WiM excels in multi-hop reasoning and summarization tasks (HoppotQA, Multihop-RAG, CWE), while performance in single-hop reasoning (SQuAD) varies by model.

| | | HotpotQA | | | MultiHop RAG | SQuAD | | | CWE | | | Average |
| --- | --- | --- | --- | --- | --- | --- | --- | --- | --- | --- | --- | --- |
| Context: | | 16k | 32k | 64k | 13-32k | 16k | 32k | 64k | 64k | | | Excl. CWE |
| Model | Pattern | Acc. | Acc. | Acc. | Acc. | Acc. | Acc. | Acc. | P | R | F1 | Acc. |
| Phi-3-small-128k-instruct | LLM | 0.47 | 0.55 | 0.48 | 0.58 | **0.81** | 0.75 | **0.79** | **0.77** | **0.77** | **0.77** | 0.52 |
| | RAG | 0.55 | 0.56 | 0.50 | 0.70 | **0.81** | 0.78 | 0.79 | 0.65 | 0.64 | 0.65 | **0.58** |
| | WiM | **0.66** | **0.64** | **0.56** | **0.77** | 0.65 | 0.74 | 0.64 | 0.70 | 0.69 | 0.69 | **0.66** |
| Qwen2-7B-Instruct | LLM | 0.62 | 0.59 | 0.39 | 0.83 | 0.81 | 0.71 | 0.57 | 0.46 | 0.46 | 0.46 | 0.61 |
| | RAG | 0.54 | 0.55 | **0.56** | 0.77 | **0.87** | **0.84** | **0.86** | 0.49 | 0.49 | 0.49 | 0.61 |
| | WiM | **0.69** | **0.66** | **0.56** | **0.92** | 0.83 | 0.80 | 0.74 | **0.69** | **0.67** | **0.68** | **0.71** |
| Meta-Llama-3.1-8B-Instruct | LLM | 0.65 | 0.64 | 0.60 | 0.85 | **0.90** | **0.92** | 0.87 | 0.22 | 0.21 | 0.22 | 0.69 |
| | RAG | 0.67 | 0.65 | 0.59 | 0.77 | 0.87 | 0.91 | **0.91** | 0.47 | 0.47 | 0.47 | 0.67 |
| | WiM | **0.77** | **0.71** | **0.73** | **0.86** | 0.88 | 0.85 | 0.82 | **0.94** | **0.93** | **0.93** | **0.77** |
| Phi-3-medium-128k-instruct | LLM | 0.57 | 0.53 | 0.48 | 0.80 | 0.84 | 0.72 | 0.70 | **0.91** | **0.91** | **0.91** | 0.60 |
| | RAG | 0.50 | 0.55 | 0.51 | 0.78 | **0.86** | **0.82** | **0.83** | **0.91** | **0.91** | **0.91** | 0.59 |
| | WiM | **0.63** | **0.67** | **0.57** | **0.93** | 0.81 | 0.80 | 0.77 | 0.90 | 0.90 | 0.90 | **0.70** |
| Palmyra-4-Chat-128K | LLM | **0.70** | 0.60 | 0.57 | 0.85 | **0.84** | 0.76 | 0.73 | 0.76 | 0.77 | 0.76 | 0.68 |
| | RAG | 0.59 | 0.54 | 0.55 | 0.78 | 0.74 | 0.70 | 0.69 | **0.80** | **0.80** | **0.80** | 0.62 |
| | WiM | 0.69 | **0.63** | **0.66** | **0.86** | 0.78 | **0.77** | **0.74** | 0.77 | 0.77 | 0.77 | **0.71** |
| Meta-Llama-3.1-70B-Instruct | LLM | **0.80** | 0.74 | 0.70 | **0.91** | **0.93** | 0.85 | 0.87 | 0.37 | 0.36 | 0.36 | **0.79** |
| | RAG | 0.73 | 0.72 | 0.63 | 0.80 | 0.90 | **0.92** | **0.95** | 0.66 | 0.65 | 0.66 | 0.72 |
| | WiM | 0.79 | **0.76** | **0.71** | 0.89 | 0.90 | 0.90 | 0.82 | **1.00** | **1.00** | **1.00** | 0.79 |
| Qwen2-72B-Instruct | LLM | 0.75 | 0.72 | 0.57 | **0.88** | 0.91 | 0.78 | 0.76 | 0.42 | 0.36 | 0.39 | 0.73 |
| | RAG | 0.70 | 0.66 | **0.70** | 0.80 | **0.92** | 0.87 | **0.91** | 0.75 | 0.75 | 0.75 | 0.72 |
| | WiM | **0.80** | **0.79** | **0.70** | **0.88** | 0.88 | **0.88** | 0.87 | **0.98** | **0.98** | **0.98** | **0.79** |
| **Average** | LLM | 0.65 | 0.62 | 0.54 | 0.81 | **0.86** | 0.78 | 0.76 | 0.56 | 0.55 | 0.55 | 0.66 |
| | RAG | 0.61 | 0.60 | 0.58 | 0.77 | 0.85 | **0.83** | **0.85** | 0.68 | 0.67 | 0.68 | 0.64 |
| | WiM | **0.72** | **0.69** | **0.64** | **0.87** | 0.82 | 0.82 | 0.77 | **0.85** | **0.85** | **0.85** | **0.73** |

By looking at different length variants of HotpotQA ($16k$, $32k$, $64k$) we see that all patterns lose accuracy as we add more context (LLM: from 0.65 to 0.54, RAG: from 0.61 to 0.58, WiM: from 0.72 to 0.64). This observation aligns with the notion that extending the context length in models degrades the performance of complex reasoning tasks. However, using WiM allows us to maintain almost the same accuracy for $64k$ as the LLM achieves on $16k$.

## 4.2 NEEDLE RETRIEVAL AND SINGLE-HOP QUESTION ANSWERING

Analysis of the SQuAD benchmark results shows that all scores are distributed across similar values with a slight preference for RAG. WiM prompting increase verbosity of LLMs, which is distracting for SQuAD expecting short answers. Nevertheless, we see that replacing an LLM with the WiM pattern consistently improves accuracy in SQuAD by $2\% - 17\%$ for Qwen2-7B-Instruct, whereas LLM is a preferred inference pattern for $16k$ context window for 4 out of 7 tested models.

Unsurprisingly, RAG emerges as the most optimal pattern for six out of seven evaluated models when extending the context length to 64k tokens in SQuAD. Indeed, for single-hop reasoning tasks, if the filtering process is successful (here we approximate the retriever by an LLM classifier), the challenge is reduced to a trivial task of retrieving a needle from a context window of 4096 tokens. However, this assumption in the RAG setup is overly optimistic because the LLMs used in our experiment are at least $7B$ in model parameters, and such large models are not typically used as retrievers. In practical scenarios, one might expect the results to be even more favorable for both LLM and WiM compared to RAG.

## 4.3 AGGREGATION

The pattern across the data indicates that WiM either matches or substantially boosts the aggregation skills of off-the-shelf models, giving an LLM on average a $30\%$ increase in F1-score for the CWE benchmark, and outperforming RAG by $17\%$.

Table 5: **Ablation: Filtering Margins and Content Compression** Removing irrelevant margins improves results for most models across HotpotQA, Multihop-RAG, and SQuAD benchmarks. Using both margins and full context generally boosts scores, despite performance drops in models with longer inputs. Results aggregated over HotpotQA, Multihop-RAG, and SQuAD benchmarks.

| Model | Margin Filter | | Content Compression | | |
| --- | --- | --- | --- | --- | --- |
| | Unfiltered | Filtered (WiM) | Only Margins | Only Context | Both (WiM) |
| Phi-3-small-128k-instruct | 0.54 | **0.58** | **0.60** | 0.55 | 0.58 |
| Qwen2-7B-Instruct | 0.63 | **0.65** | 0.62 | 0.56 | **0.65** |
| Meta-Llama-3.1-8B-Instruct | **0.70** | **0.70** | 0.68 | 0.68 | **0.70** |
| Phi-3-medium-128k-instruct | 0.64 | **0.65** | 0.57 | 0.58 | **0.65** |
| Palmyra-4-Chat-128K | 0.55 | **0.64** | 0.53 | 0.63 | **0.64** |
| Meta-Llama-3.1-70B-Instruct | **0.73** | 0.72 | **0.72** | **0.72** | 0.72 |
| Qwen2-72B-Instruct | 0.71 | **0.72** | **0.72** | 0.67 | **0.72** |

We observe that CWE results can be grouped into four classes, which surprisingly tend to align more with the model families than with the model sizes. Models like Meta-Llama-3.1-8B-Instruct and Meta-Llama-3.1-70B-Instruct achieve a remarkably significant boost in F1-score when using WiM across all context lengths, reaching up to 72% compared to the LLM baseline. Conversely, models like Phi-3-small-128k-instruct and Phi-3-medium-128k-instruct consistently prefer the vanilla LLM inference. Meanwhile, Qwen2-7B-Instruct and Qwen2-72B-Instruct point to WiM as the most optimal pattern, showing a moderate improvement ranging from 22% to 59%. RAG is preferred only by Palmyra-4-Chat-128K, which outperforms the rest by 3% − 4%.

After reviewing all comments provided by GPT-4-turbo during the evaluation of each data point, we observe that models often resort to writing Python code to solve the problem (18.5% of all answers), leading to incorrect or generic answers (resulting in on average 20% drop in F1-score).

## 5 ABLATION STUDY

### 5.1 NO MARGINS FILTERING

In this experiment, we excluded the margins classifier from the WiM pipeline, which resulted in all extractive summaries being appended directly to the context.

Table 5 presents the accuracy scores aggregated over three benchmarks: HotpotQA, MultiHop-RAG, SQuAD. Including all margins in the context decreases the accuracy by up to 8% compared to the original WiM pipeline. This effect is analogous to negative instruction manipulation, akin to telling the model to "forget all previous instructions". Ultimately, filtering margins—especially when combined with the margin generation step—not only saves computation by allowing irrelevant margins to be dropped, but also improves the overall performance.

### 5.2 REPLACING THE CONTENT BY MARGINS

An effective approach to reduce computational demands is to eliminate the KV cache in the final step, relying only on extracted positive margins. This method compresses the long context document based on the query. Although retaining the full context may capture answers better, increased input length has been shown to reduce model performance.

Table 5 also presents aggregated results for HotpotQA, MultiHop-RAG and SQuAD, demonstrating that incorporating both margins and the complete document consistently maximized the performance for almost all evaluated models, except for Meta-Phi-3-small-128k-instruct. Employing a query-based extractive summary—specifically, using only the content from margins—gave mixed results across all models; e.g., Meta-Llama-3.1-70B-Instruct scores were consistent across all metrics (0.72), while Palmyra-4-Chat-128K scores saw a decrease from 0.64 to 0.53. On the other hand, the model Phi-3-small-128k-instruct experienced an increase from 0.58 to 0.6. We hypothesize that these outcomes might vary depending on the specific task at hand. It is plausible that for tasks such

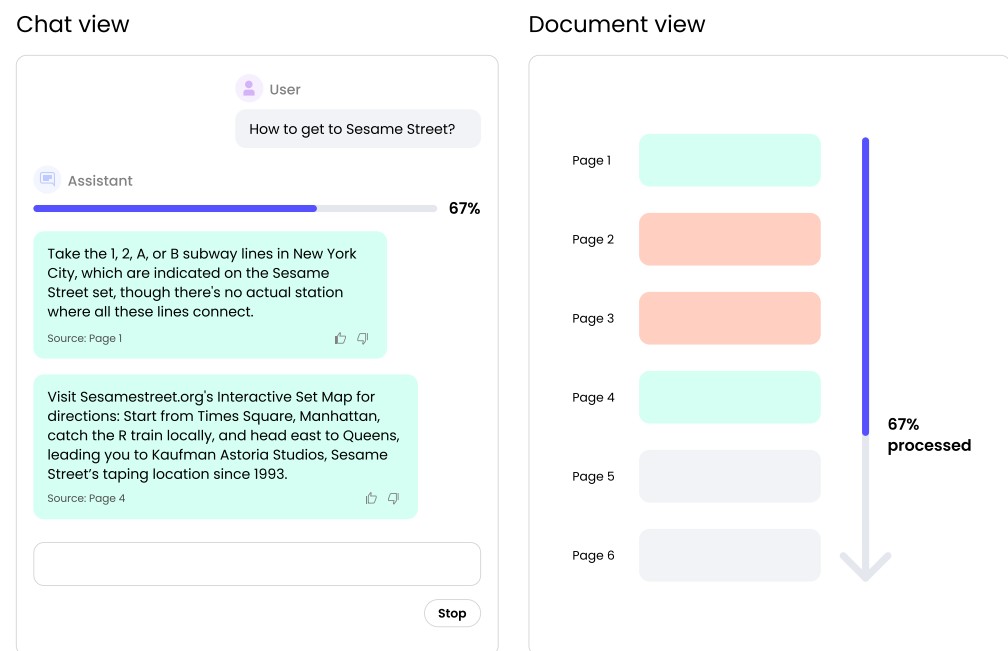

Figure 3: **WiM interactive retrieval design.** The right side displays the document view, showing processed segments, which can be labeled for relevance by the LLM classifier. The left side features a chat view with a progress bar for segment processing. Users can interact by approving or rejecting margins, and these interactions influence the final response. Each margin corresponds to a specific document segment.

as filtering and improving recall (i.e., when models are fine-tuned for margin generation and classification tasks), using margins could prove beneficial as the filtered-out content would be entirely irrelevant.

## 6 INTERACTIVE RETRIEVAL

**Explainability**   The design principles behind WiM focus not just on enhancing final benchmark performance, but also on improving the user experience. By presenting intermediate computation steps, WiM renders the decision-making process of LLMs transparent. This clarity in the model's reasoning process aids not only in debugging but also provides insights that are crucial for both end-users and developers, ensuring outputs that are both understandable and reliable.

**Latency**   Handling long documents can degrade user experience due to significant latency, as the model becomes unresponsive during processing, which can take minutes without clear indications of wait time. Our design addresses this by providing relevant information during processing and by segment-wise processing that incorporates a progress bar, thus reducing the initial response latency.

**Early exit**   WiM also offers an "early exit" option, allowing users to stop the computation if they find a satisfactory answer within any of the displayed margins. For example, in single-hop question-answering scenarios, once the answer is found in a particular section, there is no need to process further.

**Human in the Loop**   Users have the ability to improve the decision-making process by adding labels to the margins displayed in WiM. In this design, the final answer considers both the full context and the user-labeled margins. Users can evaluate and label the streamed margins (e.g., with a thumbs up or down), and these inputs could be reintegrated into the final decision-making step. The proposed design, including this feedback loop, is illustrated in Figure 3.

## 7 RELATED WORK

**External Memory and Retrieval Methods**    Memory augmentation in Large Language Models (LLMs) involves integrating external memory banks, such as k-nearest neighbor (k-NN) models, to use textual similarities for generating context-aware completions (Khandelwal et al., 2020). These k-NN based LLMs excel in managing irregular patterns and factual data (Daelemans et al., 1999). Additionally, approaches like Retrieval-Augmented Generation (RAG) (Lewis et al., 2020) and "Entities as Experts" (Févry et al., 2020) link LLMs with external data sources—ranging from structured knowledge graphs (Liu et al., 2022) to learned entity embeddings. Such methods allow LLMs to access and utilize external information to enhance response accuracy and relevance.

**Scratchpad Mechanisms**    A method for intermediate computation in LLMs involves the use of "scratchpads" or CoT (Wei et al., 2022) as a method for improving handling of sustained multi-step computations. Adopted from the findings of "Show Your Work: Scratchpads for Intermediate Computation with Language Models" (Nye et al., 2022) this method enables LLMs to show their logic step-by-step, similar to a human using paper to jot down interim calculations. By training Transformers to sequentially output the results of intermediate steps rather than only final answers, LLMs demonstrate enhanced performance on complex tasks that go beyond single-step reasoning, such as long addition and program execution. This method not only helps the model maintain and extend context dynamically but also aids in debugging and understanding model decisions (Austin et al., 2021). Further studies into length generalization have demonstrated that traditional fine-tuning techniques on tasks requiring such generalizations often encounter significant limitations (Anil et al., 2022). By integrating scratchpad-like methodologies, these language models can achieve a notable improvement in handling progressively longer text spans. This enhancement proves particularly valuable for challenges such as theorem proving and extensive text synthesis. Here, the in-context learning combined with the sequential output of computed steps substantially bolsters task accuracy and model robustness (Chen et al., 2021; Wu et al., 2021).

**Context Aggregation**    Efficiency in context aggregation for LLMs have evolved with methods like Fusion-in-Decoder (FiD) and Map Reduce. FiD, used in models such as T5 and BART, consolidates contextual embeddings via encoder and decoder components to ensure comprehensive information integration (Ivgi et al., 2023; Izacard & Grave, 2021). Conversely, LangChain's Map Reduce processes segments in parallel to quickly synthesize responses into a refined final output (Chase, 2022). Parallel Context Windows (PCW) and Naive Bayes Context Extension (NBCE) further enhance handling of extended contexts by partitioning these into smaller segments for efficient parallel processing, optimizing both processing speed and response relevance (Su et al., 2024; Ratner et al., 2023).

## 8 CONCLUSION

In this paper, we have introduced a new inference pattern called Writing in the Margins (WiM), which leverages chunked prefill to add only a marginal computational cost, emulating the human behavior of making notes in the margins. We demonstrated that this inference pattern significantly boosts the performance of off-the-shelf models across various long-context, retrieval-oriented tasks, including multi-hop reasoning (by 7.5% in HotpotQA, MultiHop-RAG), and aggregation (by 30.0% in CWE). Remarkably, this method does not require finetuning and is compatible with any transformer model.

Additionally, our approach enhances end-user experience by making context processing more transparent. By streaming "margins" that influence final predictions, our design supports early engagement. WiM differs from traditional long-context methods by allowing immediate streaming of relevant margins after segment processing, improving latency and reducing computational demands through an early exit strategy. This feature facilitates human-in-the-loop involvement in LLM decision-making, boosting interaction and intervention opportunities.

Our innovation decouples training and inference, building on ideas like CoT. By merging KV cache management with targeted prompting strategies, our approach complements existing prompt-based methods, aiming to initiate research into KV cache-aware prompting. This could improve LLM reasoning abilities and add a layer of interpretability.

**Reproducibility Statement**   We have made the code for reproducing our results available through the HuggingFace transformer library at an <anonymised URL>. The evaluation data can be obtained from the HuggingFace Hub under MultiHop-RAG[3], or it can be generated using the RULER code[4] for datasets such as SQuAD, HotpotQA, and CWE, with the specific parameters detailed in our paper. Both sources are provided under a permissive license. Furthermore, we have also disclosed the inference parameters and prompts needed to replicate our results.

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

# A    APPENDIX - PSEUDOCODE FOR CHUNKEDPREFILL AND WRITING IN THE MARGINS ALGORITHMS

---

**Algorithm 1:** Inference with Chunked Prefill

---

**Input**  : system_message (string)
          context (string)
          instruction (string)
          llm (object)
**Output:** output (string)
1  context ← system_message + context;
2  segments ← `split` (context) ;
3  past_key_value ← [];
4  **for** segment ∈ segments **do**
      // add the segment to the KV cache
5     | `prefill` (llm, past_key_value, segment) ;
6  **end**
7  output ← `generate` (llm, past_key_value, instruction) ;
8  **return** output

---

**Algorithm 2:** Writing in the Margins

---

**Input**  : system_message (string)
          context (string)
          instruction (string)
          extractive_summary_prompt (string)
          classification_prompt (string)
          llm (object)
**Output:** output (string)
1  context ← system_message + context;
2  segments ← `split` (context) ;
3  past_key_value ← [];
4  positive_margins ← [];
5  **for** segment ∈ segments **do**
      // add the segment to the KV cache
6     | `prefill` (llm, past_key_value, segment) ;
      // generate using the content of the KV cache and then discard any
      // tokens added to the KV cache by the prompt and the generated tokens
7     | margin ← `generate` (llm, past_key_value, extractive_summary_prompt) ;
8     | classification_input ← `format` (classification_prompt, margin, instruction) ;
9  **end**
   // do not use any past KV cache to classify
10 classification_result ← `generate` (llm, *NULL*, classification_input) ;
11 **if** classification_result = *true* **then**
12    | `append` (positive_margins, margin)
13 **end**
14 all_positive_margins ← `concatenate` (positive_margins) ;
15 `prefill` (llm, past_key_value, all_positive_margins) ;
16 output ← `generate` (llm, past_key_value, instruction) ;
17 **return** output

---

| | Hello | my | name | is | John | This | is | a | dog |
|---|---|---|---|---|---|---|---|---|---|
| Hello | 1 | 0 | 0 | 0 | 0 | 0 | 0 | 0 | 0 |
| my | 1 | 1 | 0 | 0 | 0 | 0 | 0 | 0 | 0 |
| name | 1 | 1 | 1 | 0 | 0 | 0 | 0 | 0 | 0 |
| is | 1 | 1 | 1 | 1 | 0 | 0 | 0 | 0 | 0 |
| John | 1 | 1 | 1 | 1 | 1 | 0 | 0 | 0 | 0 |
| This | 0 | 0 | 0 | 0 | 0 | 1 | 0 | 0 | 0 |
| is | 0 | 0 | 0 | 0 | 0 | 1 | 1 | 0 | 0 |
| a | 0 | 0 | 0 | 0 | 0 | 1 | 1 | 1 | 0 |
| dog | 0 | 0 | 0 | 0 | 0 | 1 | 1 | 1 | 1 |

Figure 4: **Sequence packing.** Sequence packing allows to pack multiple unrelated documents in the same sequence. By adjusting the attention mask, we can avoid cross-contamination. This speeds up training time by reducing the number of padding tokens. A similar technique can also be used to inference from multiple prompts using the same sequence.

## B    APPENDIX - DECOUPLING EXTRACTION AND CLASSIFICATION

Writing in the margins generates supplemental information by leveraging a partially prefilled KV cache. Each subsequent segment $c$ in the KV cache can be used to generate an annotation, known as a "margin note". To avoid providing the model with all the margins, we ask the model to generate the first token corresponding to the margin classes: relevant vs. irrelevant. In this section, we explore the possibility of decoupling the extraction and classification steps, which will allow for using separate prompting strategies. This separation might further boost the performance of the WiM pattern. We demonstrate that one can use the same instance of the model to perform both the computation of the margins and their classification.

In a naive implementation of such overlapped computation, the user may treat the classification request as an additional sequence and batch it with the prefilling request; this approach would require a very large number of padding tokens to align the two sequences. A more computationally efficient solution is to pack the classification request into the same sequence used to prefill the context and adjust the attention mask accordingly. An example of such a mask is provided in Figure 4. This technique is utilized during the pre-training of language models to reduce the number of padding tokens.

The first request to the language model would only contain the first segment $c_1$ and the additional extractive instruction $I_A$ (the"extractive summary prompt"). The attention mask at this point is provided in Figure 5 and Figure 6. This would generate the first margin $M_0$. After generating $M_0$, the instruction prompt $I_A$ and all the subsequent tokens generated in $M_0$ can be removed from the KV cache, leaving the KV cache only with $c_1$. In order to not grow or shrink a dynamically allocated KV cache, it is possible to use a static KV cache, as the number of total tokens in each segment,

| | | $c_1$ | | | | $I_A$ | | PAD | | |
|---|---|---|---|---|---|---|---|---|---|---|
| | | K1 | K2 | K3 | K4 | K5 | K6 | P | P | ... |
| $c_1$ | Q1 | 1 | 0 | 0 | 0 | 0 | 0 | 0 | 0 | 0 |
| | Q2 | 1 | 1 | 0 | 0 | 0 | 0 | 0 | 0 | 0 |
| | Q3 | 1 | 1 | 1 | 0 | 0 | 0 | 0 | 0 | 0 |
| | Q4 | 1 | 1 | 1 | 1 | 0 | 0 | 0 | 0 | 0 |
| $I_A$ | Q5 | 1 | 1 | 1 | 1 | 1 | 0 | 0 | 0 | 0 |
| | Q6 | 1 | 1 | 1 | 1 | 1 | 1 | 0 | 0 | 0 |

Figure 5: **Prefilling of the first segment $c_1$ along with the extractive instruction $I_A$.** Padding tokens are shown for clarity in case of a statically allocated KV cache, but they do not needed to be attended to or used in the KV sequence when calculating the attention. The KV sequence should be a *slice* of the KV tensor that includes only non-padding tokens.

| | | $c_1$ | | | | $I_A$ | | $M_0$ | PAD | |
|---|---|---|---|---|---|---|---|---|---|---|
| | | K1 | K2 | K3 | K4 | K5 | K6 | K7 | P | ... |
| $M_0$ | Q7 | 1 | 1 | 1 | 1 | 1 | 1 | 1 | 0 | 0 |

Figure 6: **Token generation using the prefilled KV cache.** Each generated token replaces a padding token in the KV cache.

extractive instruction and classification prompt is known in advance, so is the maximum number of tokens for each margin $M_i$ and classification result $\omega(I(M_i))$.

Having generated the first margin $M_0$, it is possible to add the second segment $c_2$ to generate the second margin $M_1$ while at the same time classifying the previously generated margin $M_0$. To do so, the KV cache is prefilled with subsequent tokens $c_2$, the extractive instruction $I_A$ and a number of padding tokens to accommodate the generated tokens of margin $M_1$. Moreover, the KV cache is also expanded by adding the classification instruction $I(M_0)$ and a number of padding tokens to accommodate the generated tokens for the classification result $\omega(I(M_0))$. The attention mask at this point is provided in Figure 7.

Autoregressive token generation of the margin $M_1$ and the classification result $\omega(I(M_0))$ can be done in parallel by projecting the last token of each sub-sequence into logits. Each generated token can then be added in place of a padding token in each subsequence to generate successive tokens. Token generation at this stage is shown in Figure 8.

By using a statically allocated KV cache and by keeping track of how many tokens are used in it, it is possible to use a partial *view* (also known as "tensor slicing") of the KV tensor without any computational overhead. It is also possible to use techniques like PagedAttention (Kwon et al., 2023) to allocate the KV cache block by block, in order to optimize the memory consumption while benefiting from a partial static allocation.

| | | $c_1$ | | | | $c_2$ | | | | $I_A$ | | PAD | | $I(M_0)$ | | | PAD | | |
|---|---|---|---|---|---|---|---|---|---|---|---|---|---|---|---|---|---|---|---|
| | | K1 | K2 | K3 | K4 | K5 | K6 | K7 | K8 | K9 | K10 | P | ... | K21 | K22 | K23 | P | P | ... |
| $c_2$ | Q5 | 1 | 1 | 1 | 1 | 1 | 0 | 0 | 0 | 0 | 0 | 0 | 0 | 0 | 0 | 0 | 0 | 0 | 0 |
| | Q6 | 1 | 1 | 1 | 1 | 1 | 1 | 0 | 0 | 0 | 0 | 0 | 0 | 0 | 0 | 0 | 0 | 0 | 0 |
| | Q7 | 1 | 1 | 1 | 1 | 1 | 1 | 1 | 0 | 0 | 0 | 0 | 0 | 0 | 0 | 0 | 0 | 0 | 0 |
| | Q8 | 1 | 1 | 1 | 1 | 1 | 1 | 1 | 1 | 0 | 0 | 0 | 0 | 0 | 0 | 0 | 0 | 0 | 0 |
| $I_A$ | Q9 | 1 | 1 | 1 | 1 | 1 | 1 | 1 | 1 | 1 | 0 | 0 | 0 | 0 | 0 | 0 | 0 | 0 | 0 |
| | Q10 | 1 | 1 | 1 | 1 | 1 | 1 | 1 | 1 | 1 | 1 | 0 | 0 | 0 | 0 | 0 | 0 | 0 | 0 |
| $I(M_0)$ | Q21 | 0 | 0 | 0 | 0 | 0 | 0 | 0 | 0 | 0 | 0 | 0 | 0 | 1 | 0 | 0 | 0 | 0 | 0 |
| | Q22 | 0 | 0 | 0 | 0 | 0 | 0 | 0 | 0 | 0 | 0 | 0 | 0 | 1 | 1 | 0 | 0 | 0 | 0 |
| | Q23 | 0 | 0 | 0 | 0 | 0 | 0 | 0 | 0 | 0 | 0 | 0 | 0 | 1 | 1 | 1 | 0 | 0 | 0 |

Figure 7: **Prefilling of the second segment $c_1$ along with the extractive instruction $I_A$.** In this case the padding tokens between $I_A$ and $I(M_0)$ must be included in the KV sequence when calculating the attention to retain the memory continuity of the tensor, but the terminal padding tokens need not to. Each token in the second segment $c_2$ needs to attend all tokens in the first segment $c_1$. The classification prompt $I(M_0)$ be considered a completely separate document in the same sequence as prefilling.

| | | $c_1$ | | | | $c_2$ | | | | $I_A$ | | $M_1$ | PAD | $I(M_0)$ | | | $\omega\big(I(M_0)\big)$ | PAD | |
|---|---|---|---|---|---|---|---|---|---|---|---|---|---|---|---|---|---|---|---|
| | | K1 | K2 | K3 | K4 | K5 | K6 | K7 | K8 | K9 | K10 | K11 | ... | K21 | K22 | K23 | K24 | P | ... |
| $M_1$ | Q11 | 1 | 1 | 1 | 1 | 1 | 1 | 1 | 1 | 1 | 1 | 1 | 0 | 0 | 0 | 0 | 0 | 0 | 0 |
| $\omega\big(I(M_0)\big)$ | Q24 | 0 | 0 | 0 | 0 | 0 | 0 | 0 | 0 | 0 | 0 | 0 | 0 | 1 | 1 | 1 | 1 | 0 | 0 |

Figure 8: **Parallel token generation of the margin $M_1$ and the classification result $\omega(I(M_0))$.** Each generated token replaces a padding token in its specific subsequence.

## C  APPENDIX - PROMPTING

For all benchmarks, we respected their original formulation. In all cases, the prompt strategy for the Long Context LLM baseline could be expressed as:

```
{system_message}
```text
{context}
```

{instruction}
{query}
```

Where `system_message` and `instruction` were usually the task instructions split into two parts and appended before and after the main context respectively.

In the RAG approach, we used the original prompt but replaced `context` with all relevant segments concatenated by a newline sign.

In WiM inference, all constructed prompts shared the common prefix:

```
{system_message}
```text
{context}
```
```

This was necessary for the efficient reuse of the KV cache. To ensure that predictions were comparable, we manually identified a promising prompt for the margin generation and final prediction steps for all evaluated models.

### C.1  MARGIN GENERATION

For each intermediate context $\texttt{context}_i = \Sigma_1^i c_i$ and instruction $I$, we used the following extractive summary prompt $I_A$ to generate a margin note $M_i$:

```
I_A = """
{system_message}
```text
{context_i}
```
Copy over all context relevant to the query: {query}
Provide the answer in the format: <YES/NO>#<Relevant context>.
Here are rules:
- If you don't know how to answer the query - start your answer with NO#
- If the text is not related to the query - start your answer with NO#
- If you can extract relevant information - start your answer with YES#
- If the text does not mention the person by name - start your answer
    ↪ with NO#
Example answers:
- YES#Western philosophy originated in Ancient Greece in the 6th century
    ↪ BCE with the pre-Socratics.
- NO#No relevant context.
"""
```

In our experiments, the margin generation step was combined with the classification step; the first token generated was a class label. We conditioned the generation of a margin based on the first token; i.e., we continued the generation only if the first token was YES. Additionally, the prompt included an explanation designed to enforce specific formatting and to prevent the model from inserting comments before delivering its judgment.

In Appendix B, we explore the possibility of decoupling margin generation and classification prompts while using the same instance of the model.

### C.2 FINAL WIM PROMPT WITH ACCUMULATED MARGINS

We used two variants of the prompt, depending on the number of retrieved margins.

#### C.2.1 SINGLE MARGIN

```
{system_message}
```text
{context}
```
I asked my assistant to read and analyse the above content page by page
    ↪ to help you complete this task. This is a margin note left on the
    ↪ last page:
```text
QUERY: {query}
ANSWER: {M_i}
```
Read again the note(s) and the provided content, take a deep breath and
    ↪ answer the query.
{instruction}
{query}
```

#### C.2.2 MULTIPLE MARGINS

```
{system_message}
```text
{context}
```
I asked my assistant to read and analyse the above content page by page
    ↪ to help you complete this task. Those are margin notes left on each
    ↪  page:
```text
Page 0:
QUERY: {query}
ANSWER: {M_i}
```

```
Page 1:
QUERY: {query}
ANSWER: {M_j}
...
```

Read again the note(s) and the provided content, take a deep breath and
    ↪ answer the query.
{instruction}
{query}
```

We replaced the term "segment" with "page" to more closely replicate the human practice of writing in the margins. In our experiments, there was no relationship between the order of the segments and the page numbers; this is left as an optional implementation detail.

### C.3 PROMPT USED WITH ACCURACY METRIC FOR SQUAD, HTOPOTQA AND MULTIHOP-RAG

```
Evaluate the following exam answer. I will provide you with the query,
    ↪ target answer(s) and the answer provided by the student.
The student's answer does not need to preserve the casing of the target
    ↪ answers, and slight variations in phrasing are acceptable, provided
    ↪  the meaning remains correct.
Provide the answer in the format: <YES/NO>#<Explanation>.

Here are the rules:
- If the student's answer is correct - start your answer with YES#
- If the student's answer is wrong or it is missing - start your answer
    ↪ with NO#

Example answers:

QUERY: As of 2016, about what percentage of adults aged 18 years or older
    ↪  were overweight?
TARGET: 40%, forty percent
ANSWER: forty percent
YES#The student's answer is correct.

QUERY: What is the value of p in 24 = 2p?
TARGET: 12, 12.0
ANSWER: five
NO#The student's answer is wrong.

QUERY: What is the 'Lotus principle'?
TARGET: The so-called Lotus principle is that 'restrictions upon the
    ↪ independence of States cannot therefore be presumed
ANSWER: The Lotus principle is a horticultural technique developed in
    ↪ ancient Egypt for cross-pollinating lotus flowers with roses to
    ↪ create fragrant, floating gardens.
NO#No, the student's explanation is wrong.

QUERY: {query}
TARGET: {target}
ANSWER: {answer}
```

### C.4 PROMPT USED WITH F1 METRIC FOR CWE

```
Evaluate the following exam answer. The student's task was to identify
    ↪ the most common words in text. I will provide you with the target
    ↪ answer(s) - 10 words - and the student's response.
The student's answer does not need to preserve the casing of the target
    ↪ words.
Provide your assessment in the format: <Positive>#<Negative>#<Explanation
    ↪ >.
```

Here are the rules:
- <Positive> denotes the number of common words that the student has
  ↪ correctly identified (i.e., those included in the target list).
- <Negative> represents the number of words the student has identified as
  ↪ common but are incorrect (i.e., not included in the target list).
- If the student's answer is non-relevant, simply return: 0#10#The answer
  ↪ is garbage.
- If the student's answer includes Python code, mention "python" in your
  ↪ explanation.
- If the student provides the full answer twice, consider only the first
  ↪ submission.

Example answers:

TARGET: jealousy, access, chase, recess, head, animal, capon, hops,
  ↪ unequaled, mozzarella
ANSWER: Based on the provided content and the margin notes, the 10 most
  ↪ common words in the input are:\n\n1. chase (100 occurrences)\n2.
  ↪ unequaled (99 occurrences)\n3. access (98 occurrences)\n4. hops (97
  ↪ occurrences)\n5. animal (95 occurrences)\n6. jealousy (96
  ↪ occurrences)\n7. mozzarella (94 occurrences)\n8. recess (93
  ↪ occurrences)\n9. capon (91 occurrences)\n10. oxygen (91 occurrences
  ↪ ),
9#1#The student's answer contains 9 correct words out of 10 given.

TARGET: seeder, workout, wafer, chronograph, liability, hovercraft,
  ↪ nestling, alloy, netball, addicted
ANSWER: To find the 10 most common words in the given input, we need to
  ↪ count the occurrences of each word and then sort them in descending
  ↪ order. Here are the 10 most common words and their counts:\n\n1.
  ↪ \"floodplain\" - 25 occurrences\n2. \"acetate\" - 24 occurrences\n3
  ↪ . \"nondisclosure\" - 23 occurrences\n4. \"caterpillar\" - 22
  ↪ occurrences\n5. \"moai\" - 21 occurrences\n6. \"video\" - 20
  ↪ occurrences\n7. \"bacterium\" - 19 occurrences\n8. \"frog\" - 19
  ↪ occurrences\n9. \"carnivore\" - 18 occurrences\n10. \"cattle\" - 18
  ↪ occurrences\n\nNote: The input provided is a long list of words,
  ↪ and the above list is based on the most frequent words found in the
  ↪ list. The actual list may vary depending on the specific input.
0#10#The student's answer provides only a generic answer and fails to
  ↪ solve the problem.

TARGET: jealousy, access, chase, recess, head, animal, capon, hops,
  ↪ unequaled, mozzarella
ANSWER: To find the 10 most common words in the provided input, we can
  ↪ use a Python script that reads the text, splits it into words,
  ↪ counts the occurrences of each word, and then prints the top 10
  ↪ most common words along with their counts. Here's a Python script
  ↪ that accomplishes this:\n\npython\nfrom collections import Counter\
  ↪ n\n# Read the text from the input\nwith open('input.txt', 'r') as
  ↪ file:\n text = file.read()\n\n# Split the text into words\nwords =
  ↪ text.split()\n\n# Count the occurrences of each word\nword_counts =
  ↪ Counter(words)\n\n# Get the 10 most common words\
  ↪ nmost_common_words = word_counts.most_common(10)\n\n# Print the 10
  ↪ most common words and their counts\nfor word, count in
  ↪ most_common_words:\n print(word, count) The provided text\ntext =
  ↪ 1.jealousy 2. gauge 3. work 4. townhouse 5. ubiquitous 6. regulator
  ↪ 7. oxygen 8. verdict 9. war 10. verdict 11. rag 12. rag
1#11#The student's answer contains python code. One word is correct but
  ↪ it contains also other 11 incorrect words.

TARGET: {target}
ANSWER: {answer}