# OpenReview forum: "Writing in the Margins: Better Inference Patterns for Long-Context Retrieval"
_ICLR.cc/2025/Conference — Submitted to ICLR 2025_

### Official Review · Reviewer_uuU8 · 2024-10-31

**Soundness:** 3
**Presentation:** 3
**Contribution:** 3
**Rating:** 6
**Confidence:** 2

**Summary:**

The paper presents a new inference pattern called "Writing in the Margins" (WiM) that addresses the challenges of processing long input contexts in retrieval-oriented tasks. WiM leverages the chunked prefill mechanism in large language models to generate intermediate "margin notes" that summarize relevant information for the given query. These margin notes are then incorporated into the final response, leading to significant performance boosts on benchmarks like HotpotQA and Common Words Extraction compared to vanilla long-context models and retrieval-augmented approaches. The paper also discusses how WiM can enhance the transparency and interactivity of the retrieval process by providing users with real-time insights into the model's reasoning.

**Strengths:**

* The authors introduce a novel inference pattern called "Writing in the Margins" (WiM) that leverages the chunked prefill mechanism in large language models to generate intermediate "margin notes" that can guide the final prediction. This is a clever way to address the challenges of long-context processing in retrieval-oriented tasks.

* The results show that WiM can significantly boost the performance of off-the-shelf models across a range of long-context benchmarks, including multi-hop reasoning and aggregation. This demonstrates the effectiveness of the proposed approach.

**Weaknesses:**

* The experimental setup could be expanded to include more baselines, such as state-of-the-art models specifically designed for long-context processing to better assess the relative performance of WiM.

* While the results are strong, the paper could benefit from a deeper analysis of why WiM works well for some tasks (e.g., multi-hop, aggregation) but not as consistently for others (e.g., single-hop QA). Understanding the underlying mechanisms behind these performance differences would strengthen the contributions.

**Questions:**

Please refer to the "Weaknesses".

---

> ### Author Response · Authors · 2024-11-18
>
> Thank you for your time in reviewing our paper. To answer your comments/questions:
>
> In terms of adding baselines, we chose seven of the best performing off-the-shelf models that supported a context window of 128k (based on the LMSYS Chatbot Arena at the time of our study). We also only had access to a single 8xH100 node, so were unable to test some of the largest models (e.g. Llama 450B). Within these constraints, are there any specific baselines you think we should include that would make the paper stronger?
>
> As for a deeper analysis, we briefly mentioned on Line 362 that WiM tends to increase the verbosity of LLM answers, which is at odds with SQuaD that typically expects short answers. We agree this is an important observation and so will expand the analysis in this section.

---

### Official Review · Reviewer_Babj · 2024-11-03

**Soundness:** 2
**Presentation:** 3
**Contribution:** 2
**Rating:** 5
**Confidence:** 4

**Summary:**

The authors propose and investigate the usage of intermediate information (margins) for improving long-context retrieval. They compare different small and medium-size LLMs as well as a RAG like system and find improvements over these baselines in many cases.

**Strengths:**

- interesting and original idea
- comparison with several base lines
- improvements over these baselines

**Weaknesses:**

- comparison and discussion not complete, as larger models (which show less improvements) and more sophisticated RAG systems are not included

**Questions:**

1. Larger models seem to profit less from WiM (table 4), and you do not include models larger than 70B. Would models larger than 70B still see improvements with WiM? Can you discuss this in more detail?
2. RAG is best with SQuAD in many cases, and almost always better than WiM. You argue that with multihop Q&A this is no longer the case (as shown in table 4), but isn't this only true for your RAG implementation / approximation, and more sophisticated RAG systems would improve this score?

---

> ### Author Response · Authors · 2024-11-18
>
> Thank you for your time in reviewing our paper. To answer your comments and questions:
>
> Since we only had access to a single 8xH100 node, we were unfortunately unable to evaluate the largest models that supported 128k context windows (e.g. Llama 405B). Instead, we focused on a wide range of top models (of different sizes) according to the LMSYS Chatbot Arena in order to test WiM in a number of different scenarios. Our largest model was actually 72B parameters (Qwen2-72B-Instruct, penultimate row in Table 4) and showed similar improvements to smaller models.
>
> As for RAG, one of the reasons RAG is best with SQuAD is because WiM tends to produce more verbose output than RAG, and this is at odds with the short reference answers in SQuaD (Line 362). We also acknowledge that our RAG system is already somewhat idealised (Line 285) in that segments are selected based on a LLM classifier rather than, e.g., cosine similarity between the segment and the query, which means RAG scores may be inflated. Regardless, our intention with WiM is not to improve RAG, but to introduce a new inference pattern to improve long context inference in general.

---

### Official Review · Reviewer_ztVt · 2024-11-03

**Soundness:** 1
**Presentation:** 2
**Contribution:** 1
**Rating:** 3
**Confidence:** 2

**Summary:**

The authors present a method for improving the representation of chunked text in a prompt by computing query-specific representations (margin notes) for each chunk. They hypothesize that this expanded and query-specific text allows for more efficient and effective decoding. To test this, the authors apply their method to several baseline models across three tasks: multi-hop reasoning, single-hop retrieval, and aggregation.  Post-hoc analysis involves an ablation study.

**Strengths:**

* **Interesting approach to query-specific representation expansion.**  Bootstrapping decision-making with model information (e.g., writing the margins) is a compelling way for a model to guide itself toward a better response.
* **Focus on effectiveness and efficiency.** The authors discuss both the effectiveness of their method and how it can improve the efficiency during decoding.
* **Extensive experimentation.** Notwithstanding concerns (below), testing the approach on multiple settings and across multiple models is a rigorous way to test a model.  The authors could have improved the discussion on how performance varies and what that implies about the proposed method.

**Weaknesses:**

* **No formal statement of hypotheses.** This is perhaps implicit, but given the number of experiments, it is essential to be explicit about the precise hypotheses the experiments test.  As best I can tell, one hypothesis is that treatment with margin notes will be better than treatment with other methods (LLM and RAG baselines) across a fixed condition (e.g., length variant, task).  There are some allusions to other hypotheses (e.g., comparisons across columns), but that's less clear.  This is important because of the next point.
* **No formal hypothesis tests.** There are a lot of numbers in Table 4+.  Results in bold seem to be the max within some context.  However, it's not clear if any of these differences are (a) statistically significant and/or (b) if those tests have accounted for multiple comparisons (since these datasets are being reused...a lot).  Without this, it's difficult to understand the robustness of these results.  In order to address this, you can consult the literature on significance testing (Cohen's "Empirical Methods for Artificial Intelligence" is good; tutorials from the RecSys/information retrieval communities are also good) and correcting for multiple comparisons (see those tutorials from the RecSys/information retrieval communities).
* **Writing falls off at the end.** Starting with the ablation experiments (Section 5), the flow and writing of the paper weaken.  Why do these ablation experiments make sense?  What are the implications?  What is the argument of Section 6?  How are all of these things connected to the core hypothesis of the paper?

**Questions:**

* The main results in Table 4 present many metric values repeatedly measured using a fixed dataset and multiple algorithms.  No statistical significance tests are shown.  This severely compromises the integrity of the results. Were these tests conducted—with appropriate corrections for multiple comparisons—but not reported?

---

> ### Author Response · Authors · 2024-11-20
>
> Thank you for your time in reviewing our paper. To answer your comments/questions:
>
> We regret that you found our hypotheses unclear, but are correct in that our fundamental hypothesis is that long-context inference can be improved when the context is processed with intermediate margin notes rather than all at once. We can easily make this more explicit in the paper.
>
> The 3 experimental settings are basically:
> 1. LLM: The model is given the full, long context and asked to answer a question.
> 2. RAG: The model is given the full, long context from which it first classifies chunks as relevant/irrelevant to a question, and then second answers the question based only on the relevant chunks.
> 3. WiM: The model is given the full, long context and generates margin notes for each chunk (similar to summarising chapters in a book) which are concatenated to the full context if they are considered relevant to a question, and then the model answers the question based on the full context + margin notes.
>
> In terms of results, it feels important to clarify that each cell in Table 4 represents how well a model was able to answer a question given a context. All of these results are deterministic (temperature = 0), as the model was either able to generate an answer that matched the reference, or not. It's thus not clear to us how statistical significance tests would help in this context.
>
> Finally, the ablation studies just present variations of the main experiment. In the main experiment, WiM only makes use of margins if they are classified as relevant to the question, so we wanted to show that appending *all* margins, both relevant and irrelevant, harmed performance (Section 5.1). Similarly, Section 5.2 shows what happens if the model only has access to the margin notes when answering the question (i.e. without the full original context), which demonstrates that WiM works best 1) when irrelevant margin notes are excluded, and 2) when margin notes are appended to the full context.
> Section 6 simply lists the advantages of the WiM inference pattern in terms of a real-world use-case with human users.

---

> > ### Comment · Reviewer_ztVt · 2024-11-20
> >
> > Thanks for your reply.  I just want to be sure I understand your results.  The table's cells are average accuracies across the set of evaluation examples described in 3.1?  And you are comparing those averages in your analysis?

---

> > > ### Author Response · Authors · 2024-11-21
> > >
> > > Yep, please ask if anything is unclear!
> > >
> > > Each cell represents performance over 100 questions for each dataset/context length. So, for example, in the top left cell, we asked Phi3-small to answer 100 HotpotQA questions given up to 16k context, and it correctly answered 47/100. The cell below asks the same model to answer the same questions, except using RAG, and successfully answered 55/100. The set of 100 questions+contexts is constant for each column.
> > >
> > > Only the rightmost column and bottom row (both coloured blue) contain averages. For example, Phi3-small achieved an average performance of 52% accuracy using the LLM strategy across HotpotQA, MultiHop RAG and SQuaD, rising to 58% with RAG, and 66% with WiM.

---

> > > > ### Comment · Reviewer_ztVt · 2024-11-25
> > > >
> > > > Great!  That should be the correct information to statistically test your hypothesis.  Please see the suggestions in the original review.

---

> ### Author Response · Authors · 2024-11-26
>
> Before we consider adding significance tests to the paper, could you please explain why these stats are necessary and how they will improve our paper (and your review)?
>
> Our experimental design is based on the popular RULER and MultiHopRAG benchmarking frameworks, which carried out similar evaluations and did not require significance tests, so we would like to better understand the value these tests will add.
>
> Refs:
> MultiHopRAG: https://openreview.net/forum?id=t4eB3zYWBK#discussion
> RULER: https://openreview.net/forum?id=kIoBbc76Sy#discussion

---

> > ### Comment · Reviewer_ztVt · 2024-11-26
> >
> > The submission implicitly poses the hypothesis "long-context inference can be improved when the context is processed with intermediate margin notes rather than all at once".  A set of experiments were conducted to test this hypothesis.  Statistical significance testing provides insight into whether the observed means support the hypothesis.  In addition to the reference provided in the original review, I strongly encourage the authors to review the following for more motivation,
> >
> > * Rainio, O., Teuho, J. & Klén, R. Evaluation metrics and statistical tests for machine learning. Sci Rep 14, 6086 (2024). https://doi.org/10.1038/s41598-024-56706-x
> > * Rotem Dror, Gili Baumer, Marina Bogomolov, and Roi Reichart. Replicability analysis for natural language processing: testing significance with multiple datasets. Transactions of the Association for Computational Linguistics, 5:471--486, 2017.
> > * Rotem Dror, Gili Baumer, Segev Shlomov, and Roi Reichart. The hitchhiker's guide to testing statistical significance in natural language processing. In Proceedings of the 56th annual meeting of the Association for Computational Linguistics (volume 1: long papers), 1383--- 1392, Melbourne, Australia, July 2018. , Association for Computational Linguistics.
> > * Stefan, Angelika M. and Schönbrodt, Felix D. Big little lies: a compendium and simulation of p-hacking strategies.  R. Soc. Open Sci.10220346.  2023.
> >
> > Performing the appropriate hypothesis tests is necessary for empirical results unless the authors want the contribution to be evaluated along other dimensions.

---

### Official Review · Reviewer_Jf6u · 2024-11-04

**Soundness:** 3
**Presentation:** 4
**Contribution:** 4
**Rating:** 10
**Confidence:** 4

**Summary:**

The paper introduces a new inference methodology called "writing in margins" for long context tasks. The method builds upon the chunked prefill strategy (commonly used while dealing with long contexts to avoid the quadratic growth of memory), dividing long input contexts into manageable segments and generates "margins" or intermediate summaries for each chunk.
The margins are then classified by the same LLM as useful or not-useful and useful margins are kept as part of the context and used during decoding step.
The approach seems to significantly help LLM (especially smaller LLMs) in better accuracy during decoding.

**Strengths:**

The paper provides a number of thought provoking outcomes.
1) It showcases how adding a simple strategy of adding notes or summaries in the "margins" after each prefilled chunk can assist in improving LLM reasoning and retrieval capabilities.
2) The notes written by the LLM can potentially be used to improve explainability of the final decoded output. This is dependent on whether the question asked for the margin generation is useful. In the paper the authors ask the LLM whether the context is relevant to the query (and to provide a summary).
3) The approach is general purpose, it can be applied to any LLM without the need for finetuning which is a big win.

Overall, strong contribution.

**Weaknesses:**

1) Latency - while the authors mention that latency is slightly increased, an ablation study for this would be welcome. Since the paper uses 2 steps for each chunk - margin generation and then margin classification, you are effectively doing 2 decoding steps for the model with each chunk. This will add latency, especially if the summaries generated are long.

2) comparison against finetuned models - the paper mentions that this technique the models to perform well on tasks (long context) without the need to finetune the model (similar to rag). It would be good to include a model finetuned for the task and using the standard Long Context LLM decoding approach.

**Questions:**

1) One approach the authors could explore would be to use a separate smaller LLM as classifier. Using the base model (which can be very large) adds latency.

---

> ### Author Response · Authors · 2024-11-18
>
> Thank you for your time in reviewing our paper. To answer your comments/questions:
>
> On latency, we actually only had one extra decoding step by combining the margin generation and classification steps; i.e. if the first generated token was "NO", we skip margin generation (this also bypasses the need for a smaller LLM classifier). We discussed this in Appendix C1, but you're right that this information is important, so we will bring it into the main paper and discuss further.
>
> As for finetuning, we agree it would be informative to evaluate a finetuned model in relation to the others, but we already had a lot of content and wanted to focus on off-the-shelf models.

---

### Meta-Review · Area_Chair_vZ2F · 2024-12-21

**Metareview:**

This paper proposes a new method called Writing in the Margins (WiM) to improve inference for long-context retrieval. The strategy presented can be applied to different foundation models and requires minimal additional computation. Although a reviewer gave a very high score, after reading the reviews, discussions, and the paper itself, I think this paper has the following key weaknesses: 1) The experimental section lacks sufficient baseline comparisons, which was pointed out by multiple reviewers who provided different suggestions; 2) The current test samples are too few. Although multiple datasets are used, only 100 cases were selected from each dataset, which is insufficient to validate and demonstrate the method's effectiveness; 3) There is a lack of comparative and analytical experiments. While the authors responded during the rebuttal phase, the issues mentioned above were not addressed. Therefore, I think this paper is not ready for publication by far.

**Additional Comments On Reviewer Discussion:**

The authors responded to the reviewers' concerns during the rebuttal phase, but they were unable to address the reviewers' concerns.

---

### Decision · Program_Chairs · 2025-01-22

Reject